# "Kuchoka": Investigation of research fatigue in Mosoriot, Kenya

**Felishana Cherop[1]\*, Violet Naanyu[2], Juddy Wachira[3], Lukoye Atwoli[4]**

**1** Department of Management Science and Entrepreneurship, Moi University, Eldoret, Kenya,
**2** Department of Sociology and Anthropology, Moi University, Eldoret, Kenya, **3** Department of Mental Health and Behavioural Sciences, Moi University, Eldoret, Kenya, **4** Medical College East Africa, The Aga Khan University, Nairobi, Kenya

\* fcherop@gmail.com

## Abstract

### Background

Health research is key to the promotion of population and community health, however, conducting many research studies in a community can cause research fatigue.

### Purpose

We determined the prevalence of research fatigue and associated factors in Mosoriot, Kenya.

### Methods

We conducted a cross-sectional study in the Mosoriot community from Wednesday 28, May 2014, to Thursday 30, April 2015, involving (n = 327) community members who were randomly sampled to respond to self-administered and/or guided questionnaires. We analyzed descriptive statistics to summarise the data and used the Pearson Chi-Square test to assess the bivariate associations between the variables and conducted multivariate analyses using logistic regression models to test the hypotheses. The odds ratios and the corresponding 95% confidence limits were reported.

### Results

Research fatigue prevalence was 56.3% and the factors associated included being >35 years (OR: 2.28, 95% CL: 1.27, 4.15), being male (OR: 2.80, 95% CL:1.59, 5.00), self-employment (OR: 2.05, 95% CL: 1.06, 4.01), participating in hospital-based studies (OR: 3.59, 95% CL:1.88, 7.09), involvement in multiple researches (OR: 3.86, 95% CL:1.87, 8.27), desire to drop out of a study (OR: 11.49, 95% CL: 3.69, 43.83) and being asked personal questions (OR: 6.23, 95% CL: 3.28, 12.23).

**Data availability statement:** All relevant data are within the manuscript and its Supporting information files.

**Funding:** The author(s) received no specific funding for this work.

**Competing interests:** The authors have declared that no competing interests exist.

## Conclusion

There is a high prevalence of research fatigue (56.3%) among community members in Mosoriot who have participated in repeated research, which is associated with age, gender, income source, research setting, frequency of research engagement, desire to drop out of studies, and discomfort with questions. Addressing research fatigue would enhance ethical research conduct and promote sustained community participation in research.

## Background

The value of health research in promoting the health of communities cannot be understated because it addresses community and patients' needs and fosters a collaborative and ethical foundation for scientific discovery [1]. Engaging communities in research ensures recognition of community priorities, values, and interests that would have intrinsic ethical importance [2]. However, conducting too many research studies in one community causes research fatigue [3], raises ethical issues [4], and influences data quality [5] and future participation [6]. Research fatigue is a state where individuals or groups tire of participating in research due to high-volume research projects [7], long or sensitive interviews [3,8,9], and lack of tangible benefit, often resulting to distrust [10].

Although the global prevalence of research fatigue is still unknown [4], evidence shows substantial prevalence of 42% among injecting drug users (IDUs) in HIV studies in Karachi [11], and 52% in pooled cancer studies [5]. As a result, research fatigue has several undesired outcomes including research participants may become hostile [12–14], distrustful, or feel coerced [12,15–18], hence undermining the Belmont ethical principles and guidelines for the protection of human subjects in research including respect for persons, beneficence, and justice [4,6,12,19–24]. It also threatens scientific validity of research through selection and non-response biases and social desirability effects [4,17,18], which may compromise data quality and generalizability of findings [19,20,22].

In recognition of the presence of research fatigue in health research and insufficient data in communities that have been involved in multiple studies, there is a need to document its prevalence and ethical implications in contexts such as Kenya. This study reports findings from a community frequently engaged in multiple health research projects in Kenya.

## Methods

### Ethics statement

We obtained ethical approval from the Institutional Research Ethics Committee (IREC) in Moi Teaching and Referral Hospital (FAN: IREC 0001180) before data collection. Participants were fully informed about the purpose of the study, procedures, potential risks and benefits, and the right to withdraw from the study at any time. Participants signed a written informed consent before participation in the study.

## Study design and setting

We conducted a cross-sectional study in the Mosoriot community, within Kosirai Division, Mutwot Location, and Mosop Constituency in Nandi County in western Kenya from Wednesday 28, May 2014, to Thursday 30, April 2015. According to the health demographic survey of 2009, it was estimated that the Kosirai division has an area of approximately 195 square kilometers [76 square miles] and a population of 35,383 individuals and 6,643 households [23]. This community experiences diverse health-related challenges and various research projects have been conducted by different Ministry of Health and university institutions to address them [24–28].

## Study population

We involved (n = 327) community members who were >18 years old, had resided in the Mosoriot community for more than two years before the study, had been actively engaged in more than two research projects conducted within Mosoriot, and understood and responded to research questions. We excluded community members who had relocated from the Mosoriot community and were no longer residents at the time of the study, were not willing to participate, or consent, and community members who were sick during the study period. We recruited participants using simple random sampling across Mosoriot villages to respond to survey questionnaires. Community leaders helped circulate the study brochure in Mosoriot villages for three days before data collection and they were informed that the study would be conducted at the Chief's office which was near to them. Participants who turned up were subjected to simple random sampling. To determine the sample size, the study utilized the Cochran formula ($N = z^2 (p.q)/d2$) [29,30]. The prevalence of research fatigue was not known and this study assumed 50% because there are no available documented statistics on research fatigue in the area of study. Although the calculated sample size of (n = 384), a total of 327 questionnaires were successfully administered (response rate 85.2%). This shortfall was linked to participants' busy schedules and reliance on community leaders' mobilization efforts. All the returned questionnaires were fully completed and there were no recorded missing data. Analysis was conducted on the 327 cases.

## Data collection procedures

The fieldwork team comprised two trained research assistants stratified by gender who were first trained to understand the study procedures, ethical issues, and community engagement before data collection to ensure quality and accurate data collection supported by the authors. Participants signed a written consent before participating in the study. They were first informed of the need to consent, the study purpose, study procedures and their rights including the option to withdraw from the study at any given time to ensure full comprehension. We used a structured questionnaire to collect data between Wednesday 28, May 2014, to Thursday 30, April 2015 in the Mosoriot community at the Chief's office as the central point closest to the participants in the community. Before data collection, we created awareness about the study by circulating a study brochure describing the study in the community through the community leaders. Participants who turned up at the research site at the Chief's office were subjected to study eligibility criteria that took at least five minutes. For guided interviews, the interviewer read the questions and answers loudly to ensure participant comprehension and recorded the answers based on their response. The interviews were done in a confidential room within the Chief's office and took approximately 30 minutes. Participants were then thanked for participation and given a reimbursement of KSh.200 to cover transport costs and time in the study.

## Study tool

The instrument (S1 Text) had four major parts namely; participant socio-demographics (age, gender, marital status, sex, level of education) and Socio-economic factors included; (income, number of people in household), Types of research studies (experimental, longitudinal, and cross-sectional), research fatigue, ethical issues and biases in research. Research fatigue was assessed by asking participants to indicate if they had felt any form of fatigue exhaustion in the

multiple health research studies they have participated in, namely, physical exhaustion, emotional exhaustion, and mental exhaustion. The questions were in binary response of Yes/ No. As there were no standardized or validated tools available for measuring research fatigue in the context of health research participation, we developed a structured questionnaire to capture this construct. The tool was informed by literature on participant burden and fatigue, and piloted to ensure clarity and cultural appropriateness. Although not representing a validated scale, this instrument enabled relevant data collection on research fatigue in this population. Health research participation characteristics were assessed by asking participants about the types of health research they had participated in, time taken in the last two research studies, the length of questions, ability to understand the questions, a wish to drop out of the studies they had participated in, being asked personal and sensitive questions during the study, and if the data collection tools had repetitive questions. The questions were asked in binary responses Yes/No. Understanding of ethical issues was assessed by asking questions relating to informed consent timing, study comprehension, incentives, privacy and confidentiality, and the comfort of the research environment. The questions were in a binary response of Yes/No. Response bias was assessed by asking the respondents if they had ever answered questions in a certain way that did not represent the truth. The questions were recorded in binary responses of Yes/No. The questionnaire was developed in English, translated to Swahili, and back-translated to English to ensure consistency and piloted with four participants, reviewed by the field research team, adapted, and finalised.

### Data management and analysis

We used Statistical Package for Social Sciences (SPSS) version 21 to manage the data. There were no recorded missing data in the 327 responses. Descriptive statistics including frequencies and percentages, mean and standard deviation, median, and the corresponding interquartile range were used to summarise the data. Mental, physical, and emotional fatigue was present if a participant answered positively for the presence of each fatigue question. Overall research fatigue score was derived as the presence of either mental physical or emotional fatigue if the participants responded positively in any of them. Pearson Chi-Square test assessed the association between research fatigue and demographic characteristics, research participation characteristics, understanding of ethical issues, and biases of research. Independent variables that were significant at the bivariate level were adjusted for in multiple logistic regression models. The odds ratios were reported and the corresponding 95% confidence limits. The factors that were associated with research fatigue included socio-demographics and socio-economic factors of participants, research participation characteristics, understanding of ethical issues, and questions related to response bias.

## Results

### Socio-demographic characteristics of respondents

Table 1 presents participants' socio-demographic characteristics. We surveyed 327 participants of whom more than half 172 (52.6%) were male; at least 220 (67.3%) had either attended part of or completed secondary school. Most 248 (75.8%), were married/in a sexual relationship, 295 (90.2%) earned an income, and most 196 (73.7%), engaged in a form of self-employment. Among those with a source of income, 180 (55.0%) participants earned a low income of less than Kenya Shillings 5,000. The majority 176 (53.8%) resided in an urban/semi-urban area and most 196 (59.9%) had participated in research that took place in their household.

### Prevalence of research fatigue

Over half of 204 (62.4) participants reported spending more than 1 hour at the research site. Overall research fatigue in the Mosoriot community, was 56.3% of the sample. Most participants were physically fatigued (47.4%), emotionally fatigued (42.8%), and a few mentally fatigued (15.3%) as presented in Table 2.

**Table 1. Socio-demographic characteristics of respondents in the survey.**

| Variables | Categories | Frequency (n) N = 327 | Percent (%) |
|---|---|---|---|
| Age in years | 18-35 | 170 | 52.0 |
| | 35-55 | 146 | 96.6 |
| | 55 and above | 11 | 3.5 |
| Gender | Female | 155 | 47.4 |
| | Male | 172 | 52.6 |
| Marital Status | Single | 61 | 18.7 |
| | Married/ In sexual relationship | 248 | 75.8 |
| | Widowed/Divorced/Separated | 16 | 4.9 |
| Level of Education | Secondary and below | 220 | 67.3 |
| | College/University | 89 | 27.2 |
| Earn any Income | Yes | 295 | 90.2 |
| | No | 27 | 8.3 |
| Sources of Income | Short-Term Jobs/None | 58 | 17.8 |
| | Farming/Self-employment | 196 | 59.9 |
| | Formal Employment | 71 | 21.7 |
| Income in the last month | Less than 5000 | 180 | 55.0 |
| | 5-10,000 | 111 | 33.9 |
| | More than 10000 | 36 | 11.0 |
| Area of Residence | Rural | 150 | 45.9 |
| | Urban/Semi-Urban | 176 | 53.8 |
| Research Environment | Household | 196 | 59.9 |
| | Hospital | 107 | 32.7 |
| | Other | 24 | 7.3 |
| Distance to the place of research from home | Less than one hour | 204 | 62.4 |
| | 1-2 hours | 116 | 35.5 |
| | More than 3 hours | 7 | 2.3 |

**Table 2. Assessing prevalence of research fatigue.**

| Fatigue | n (%) |
|---|---|
| Mental | 50 (15.3) |
| Emotional | 140 (42.8) |
| Physical | 155 (47.4) |
| Overall | 184 (56.3) |

## Bivariate analyses

The bivariate analyses of research fatigue, participant demographics and research participation characteristics revealed significant associations.

**Research fatigue association with participant socio-demographic characteristics.** Our findings revealed that research fatigue was highly associated with participants aged at least 35 years and older, males, those with a secondary level of education, those who were either self-employed or in formal employment, those who participated in research studies conducted in a hospital set up compared to other settings, those who had travelled more than an hour to the

research setting, and those who participated in two or more research studies. No significant associations were established between marital status, participant income in the last month, and the place of residence as shown in Table 3.

**Research fatigue association with research participation characteristics.** Our findings further revealed significant associations between research fatigue and participants who considered time taken in the study was too long, those who had considered dropping out of studies, those who felt they were asked personal questions, and those who felt they were asked similar questions repeatedly as illustrated in Table 4.

**Research fatigue association with understanding of ethical issues.** The only ethical issue associated with research fatigue was the length of the consenting process implying that participants who spent more time during the consenting process experienced research fatigue than those who spent less time as shown in Table 5.

**Table 3. Association between research fatigue and participant socio-demographic characteristics.**

| Socio-demographic & socio-economic characteristics | Research Fatigue | |
| --- | --- | --- |
| | Yes (n = 184, 56.3%) | Chi-Square test for association (*p*-value) |
| Age group (years) | | |
| <35 | 82 (48.2%) | 0.003 |
| ≥35 | 102 (65.0%) | |
| Gender | | |
| Female | 66 (42.6%) | <0.0001 |
| Male | 118 (68.6%) | |
| Marital status | | |
| Single/Widowed/Divorced/Separated | 45 (47.9%) | 0.069 |
| Married | 139 (59.7%) | |
| Education level | | |
| Primary/None | 71 (52.2%) | |
| Secondary | 68 (66.7%) | 0.037 |
| College/University | 45 (50.6%) | |
| Income source | | |
| None/Small-scale farming/short-term jobs | 67 (47.5%) | |
| Self-employed/ Large-scale farming | 72 (62.6%) | 0.021 |
| Formal employment | 45 (63.4%) | |
| Income last month (Kshs.) | | |
| <5000 | 102 (56.7%) | 0.962 |
| ≥5000 | 82 (55.8%) | |
| Residence | | |
| Rural | 60 (60.0%) | 0.245 |
| Urban/ semi-urban | 94 (53.1%) | |
| Research setting | | |
| Household | 92 (46.9%) | |
| Hospital | 81 (75.7%) | <0.0001 |
| Other | 11 (45.8%) | |
| Travel time to research setting | | |
| <1 hour | 104 (51.0%) | 0.0179 |
| ≥1 hour | 80 (65.0%) | |
| Participant number of times in research | | |
| 1 | 18 (25.4%) | <0.0001 |
| 2 or more | 166 (64.8%) | |

**Table 4. Association between research fatigue and research participation characteristics.**

| Research participation characteristics | Research Fatigue Levels | | |
|---|---|---|---|
| | Yes (184, 56.3%) | No (%) | (p-value) |
| The time taken in research was too long | 30 (73.2) | 154 (53.8) | 0.030 |
| Length of questions were too long | 38 (64.4) | 146 (54.5) | 0.212 |
| Difficult questions asked in the study | 11 (55.0) | 173 (56.4) | 1.000 |
| Participants felt like dropping out of the study | 32 (86.5) | 152 (52.4) | 0.0002 |
| Personal questions asked in the study | 161 (71.2) | 23 (22.8) | <0.0001 |
| The language used in the study was inappropriate | 178 (57.6) | 6 (33.3) | 0.076 |
| Similar questions asked in the study repeatedly | 155 (63.3) | 29 (35.4) | <0.0001 |

**Table 5. Association between research fatigue and understanding of ethical issues.**

| Understanding of ethical issues | Research Fatigue | | |
|---|---|---|---|
| | Yes (184, 56.3%) | No | (p-value) |
| The length of the consenting process was long | 177 (58.8) | 7 (26.9) | 0.003 |
| The purpose of the research was explained | 162 (56.2) | 22 (56.4) | 1.000 |
| Incentives provided during the research | 22 (62.9) | 162 (55.5) | 0.515 |
| Influence of Incentives Provided During Research | 8 (57.1) | 176 (56.2) | 1.000 |
| Privacy observed during research | 176 (57.1) | 8 (42.1) | 0.296 |
| Comfortable environment during research | 172 (55.5) | 12 (70.6) | 0.331 |

**Research fatigue association with response bias characteristics in research.** Research fatigue was associated with participants who answered questions in a certain way to protect personal information in research and when the research process took longer than expected (Table 6).

## Multivariate analyses

**Multivariate analysis of factors associated with research fatigue.** In the multivariate analysis, we adjusted for variables that were significant in the univariate analysis including age, gender, level of education, income source, research setting, travel time to research site, participant number of times in research, the length of time in research, desire to drop out of a study, participants being asked personal and similar questions repeatedly, length of consenting time, and answering questions in a way to protect personal information. As shown in Table 7, study findings show participants

**Table 6. Association between research fatigue and response bias characteristics in research.**

| Response bias characteristics | Research Fatigue Levels | | |
|---|---|---|---|
| | Yes (184, 56.3%) | No | (p-value) |
| Answering questions that misrepresent the truth in research | 14 (50.0) | 170 (56.9) | 0.617 |
| Answering questions to protect personal information | 104 (63.0) | 80 (49.4) | 0.018 |
| Answering questions by guessing | 95 (60.9) | 89 (52.0) | 0.134 |
| Answering questions in a manner because the research took longer than expected | 117 (65.0) | 67 (45.6) | 0.001 |
| Answering questions in a way because they were unfamiliar | 19 (67.9) | 165 (55.2) | 0.274 |
| Answering questions to please organizations doing research | 35 (59.3) | 149 (55.6) | 0.706 |
| Answering questions because of a lack of interest in the study | 21 (60.0) | 163 (55.8) | 0.771 |

**Table 7. Multivariate analysis of factors associated with research fatigue.**

| Variables | Levels | Adjusted OR (95% CL) |
|---|---|---|
| Age increase in (years) ≥35 | Yes vs. No | 2.28 (1.27, 4.15) |
| Gender (Male) | Yes vs. No | 2.80 (1.59, 5.00) |
| Income source | Yes vs. No | |
| Self-employed/ Large-scale farming | Yes vs. No | 2.05 (1.06, 4.01) |
| Formal employment | Yes vs. No | 1.29 (0.62, 2.69) |
| Research setting (Hospital) | Yes vs. No | 3.59 (1.88, 7.09) |
| Participant number of times in research studies (≥2 or more) | Yes vs. No | 3.86 (1.87, 8.27) |
| Desire to drop out of the study | Yes vs. No | 11.49 (3.69, 43.83) |
| Personal questions | Yes vs. No | 6.23 (3.28, 12.23) |

OR– Odds Ratio; CL – Confidence Limits; Unadjusted – means no other covariates in the model; Adjusted – means including all the other covariates in the model.

who were at least 35 years and older were twice as likely to experience research fatigue. Being male, having a form of self-employment or formal employment, and participating in a study conducted in a hospital had increased odds of experiencing significant research fatigue. In addition, participants who were involved in health research more than two times and who desired to drop out of the study and/or were being asked personal questions had increased odds of experiencing research fatigue.

## Discussion

Our study assessed the prevalence of research fatigue in Mosoriot, Kenya, a community frequently involved in research activities. Our findings indicate that research fatigue exists in the Mosoriot community and is more pronounced among males, older individuals, and those with higher education and earning income. This suggests that individuals who are targeted for for research participation due to their perceived reliability or availability of time, may be disproportionately burdened by repeated research requests and engagements. While previous literature acknowledges that research fatigue is understudied, few studies provide prevalence estimates of participants who have experienced research fatigue [4]. For example, one study recorded a 42% prevalence of research fatigue among IDUs engaged in HIV studies, which is lower than the estimates observed in our study. In contrast, a recent systematic review and meta-analysis reported that fatigue in the general population was more prevalent among females than males [31], diverging from our findings. This divergence may reflect cultural and social roles in Mosoriot, where men may be more frequently targeted by researchers due to their perceived authority, decision-making roles, thus accumulating higher research burden. Such cultural and contextual factors underscore the importance of not generalizing fatigue patterns across settings. Ethically, this highlights the need for recruitment strategies that account for local gender dynamics to avoid overburdening specific groups. Our findings also align with a previous work showing that elderly participants may have reduced energy and concentration when repeatedly engaged in lengthy studies [8], highlighting significant ethical implications for the ethical conduct of research in heavily studied communities and the need for strategies to mitigate research fatigue.

Beyond individual characteristics, the research setting also shaped participant fatigue. Participants who engaged in research studies within hospital settings and those who participated in research more than two times experienced research fatigue. This finding is important because hospital-based recruitment may blur boundaries between clinical care and research, raising risks of perceived coercion. The contributing factors include compromised privacy and confidentiality prolonged study protocols, and less direct participant benefits [4]. A previous study similarly reported ethical challenges at the intersection of community and hospital recruitment,including confidentiality, and informed consent calling for less intrusive and non-coercive recruitment methods [32]. Additionally, asking sensitive and personal information in a less protected

environments may upset participants and possibly impact personal relationships [32]. Similarly, participants are more likely to experience research fatigue if they lack interest in parts or the entirety of research project [4,33,34]. This therefore suggests ethical measures to be adopted by researchers to control research fatigue to ensure relevant quality data collection and reduced burden of fatigue in communities, thereby protecting both participant wellbeing and data quality.

Logistical burdens also contributed to fatigue. Longer travel times to the research site and prolonged study durations were strongly associated with fatigue, risking both exhaustion and attrition. This finding concurs with a previous clinical trial that established the longer the travel distance to the clinical trial site, the more likely participants had a decreased likelihood of participation in genotype-matched trials [35]. Similarly, prolonged research duration has been linked to research fatigue [8], and intentions to drop out of the research, suggesting a need to improve the sharing of the study information, recruitment and consenting processes to enhance the overall participant research experience. A previous study established a 50% participant burden in clinical trials participation with 11% median scores for physical and 14% for psychological burdens [36], illustrating the magnitude of participant strain. Methodologically, such burdens risk skewing samples toward younger, healther, or financially stable participants, potentially biasing findings.

Our findings further revealed that participants who had spent more time during the consenting process and felt that research took longer than expected experienced research fatigue. This is concerning because consent is intended to safeguard voluntariness, yet if the process itself causes fatigue, it may undermine informed decision-making. Previous studies noted that fatigue among research participants could implicate informed consent [37], with some continuing participation despite reluctance, leading to potential coercion [38]. Participants in prior studies also expressed dissatisfaction when repeatedly asked personal questions [8], describing the process as torturous. Such experiences create negative perception and risk undermining data quality.

The most concerning finding was the desire to drop out of research that emerged as the strongest predictor of research fatigue. Affected participants were more than eleven times more likely to report fatigue, making withdrawal intent a critical signal of participant distress. Ethically, this indicates that autonomy, comfort, and wellbeing are at risk. Methodologically, consistent withdrawal intentions threaten threaten study validity through attrition and biased responses. Similar observations have been reported elsewhere, where repeated participation without benefit, rest or support has increased disengagement, or mistrust towards research initiatives [4].

Finally, we found that participants who had answered questions in a way to protect personal information reported research fatigue. This suggests that when the research process is not definite, and intrusive questions, may lead to participants to give inaccurate responses, reflecting exhaustion and disengagement [39]. Such response bias undermines the reliability and validity of study data while raising ethical concerns about uneccessariuly burdening participants for information that may not be trustworthy.

## Strengths and limitations

The strength of our study lies in the research question to address an important research fatigue gap in a community that is perceived to be over-researched and highlight the ethical implications. We used a simple random sampling method to ensure a representative selection of participants within the Mosoriot community thereby strengthening the credibility and generalizability of the study findings. Our field and research team had the relevant training and experience that allowed coherent coordination and collection of credible data. However, a few limitations must be acknowledged. To begin with, our study employed a cross-sectional study design which does not determine causal inference. Similarly, our study could suffer participant recall bias because we used a guided/self-administered questionnaire that asked community members to recall and report the types of research studies and the number of times they had been involved. To address this, our questionnaire had a clear explanation about the types of studies and examples that guided the participants to determine the type of study and the number of times they had been involved. We also had two trained research assistants who guided the participants by providing clear explanations and answering relevant questions.

In addition, our study exclusively focused on community members who had participated in more than two health research which may not fully represent the experiences of those with less research exposure and/or with a different experience in other types of research other than what was provided in the questionnaire. Similarly, these selective criteria could limit the generalizability of findings to a broader population because different participants may have varied experiences and research fatigue encounters. Furthermore, the absence of baseline data to estimate the prevalence of research fatigue accurately within the study population limits the generalizability of findings to a larger population. To address this, future studies should consider a more diverse sample with varying levels of research involvement and in-depth insights regarding research fatigue. In addition, future research can track changes in research fatigue over time in different studies/ research communities to clearly understand its dynamics and potential effects. In addition, the use of mixed methods studies promises the potential development and implementation of standardized measures for assessing research fatigue in different research contexts. A further limitation is that our measurement of research fatigue relied on a newly developed instrument, as no standardized tools or validated tools exist for this construct in the context of health research participation. While our survey instrument was piloted, translated and refined, it may not have captured all potential relevant factors and dimensions of research fatigue, and comparability with other settings is therefore limited. Future studies should build on this work to develop and validate standardized instruments for measuring research fatigue.

Furthermore, while we had a well-trained research team, they may have not had familiarity with participants or community expectations which may have likely triggered social desirability bias that is likely to influence participant responses. To mitigate this, our research team was briefed on the social-cultural expectations of the communities in western Kenya and how to ask questions. There is also a possibility that some areas within the Mosoriot community were underrepresented and/or did not receive the study information on time thereby creating unequal distribution and experiences of participants across the community. Future studies should consider wider coverage to ensure full representation to get a clear landscape of research fatigue in the community.

## Conclusion

There was a high prevalence of research fatigue (56.3%) among community members in Mosoriot who have repeatedly engaged in health research. Research fatigue was strongly associated with socio-demographic and socio-economic factors (age, gender, education, income, research setting, and travel time to research setting), research participation characteristics (number of times in research, prolonged study duration, desire to drop out of study, being asked personal and similar questions), understanding of ethical issues(length of consenting) and response bias (answering questions in a way to protect personal information). The findings highlight highlight that research fatigue a participant burden, but also an ethical issue and a source of bias that may compromise the integrity of data and the willingness of communities to engage in future studies.

## Recommendations

Researchers and research assistants should obtain appropriate training that is sensitive to ethical issues or biases particularly when conducting community research with cultural sensitivities. In addition, IRBs should consider multiple research studies in a location and research fatigue as an ethical issue that affects research engagements, develop guidelines to regulate research frequency in specific locations and carry out audits to control related or repeated studies in the same communities. The IRBs and research institutions should also establish enrollment systems to identify and manage multiple enrollments of participants in concurrent studies to prevent research fatigue and duplicate research efforts. Future studies to develop standard guidelines and models for screening and detecting research fatigue in research and communities, and develop and validate standardized measures for assessing research fatigue to enhance consistency in measurements and facilitate meaningful comparisons of prevalence rates across different research contexts.

## Supporting information

**S1 Text. Study tool.**
(DOCX)

## Acknowledgments

We sincerely appreciate the community members of Mosoriot and the health researchers at Moi University and AMPATH for participating in our study.

## Author contributions

**Conceptualization:** Felishana Cherop, Juddy Wachira, Lukoye Atwoli.

**Data curation:** Felishana Cherop, Lukoye Atwoli.

**Formal analysis:** Felishana Cherop, Violet Naanyu, Juddy Wachira, Lukoye Atwoli.

**Investigation:** Felishana Cherop.

**Methodology:** Felishana Cherop, Violet Naanyu, Juddy Wachira, Lukoye Atwoli.

**Supervision:** Juddy Wachira, Lukoye Atwoli.

**Writing – original draft:** Felishana Cherop.

**Writing – review & editing:** Felishana Cherop, Violet Naanyu, Juddy Wachira, Lukoye Atwoli.

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
