## [Decision Letter · Decision Letter 0]

2 Aug 2025

Dear Dr. Cherop,

Thank you for submitting your manuscript to PLOS ONE. After careful consideration, we feel that it has merit but does not fully meet PLOS ONE’s publication criteria as it currently stands. Therefore, we invite you to submit a revised version of the manuscript that addresses the points raised during the review process.

We look forward to receiving your revised manuscript.

Kind regards,

Meghana Ray, Ph.D.

Academic Editor

PLOS ONE

Journal Requirements:

2. Please amend your authorship list in your manuscript file to include author Felishana Jepkosgei Cherop.

3. Please amend the manuscript submission data (via Edit Submission) to include author Felishana Cherop.

Reviewers' comments:

Reviewer's Responses to Questions

**Comments to the Author**

1. Is the manuscript technically sound, and do the data support the conclusions?

Reviewer #1: Yes

Reviewer #2: Yes

2. Has the statistical analysis been performed appropriately and rigorously?

Reviewer #1: Yes

Reviewer #2: I Don't Know

3. Have the authors made all data underlying the findings in their manuscript fully available?

Reviewer #1: Yes

Reviewer #2: Yes

4. Is the manuscript presented in an intelligible fashion and written in standard English?

Reviewer #1: Yes

Reviewer #2: Yes

Reviewer #1: A paper of practical value to future health research studies.

A paper of practical value to future health research studies.

A paper of practical value to future health research studies.

A paper of practical value to future health research studies.

Reviewer #2: Reviewer's comments:

The authors have addressed "research fatigue" - an issue that is important yet generally under-explored. This is an important and well-conceived study that addresses a critical gap in community-engaged research ethics. However, the manuscript could be further improved by making the following adjustments:

Major Comments:

- Abstract

While the aim of the study was to determine prevalence of research fatigue and associated factors, the conclusion portion of the abstract did not report on prevalence.

- Background

The background is informative but overly long. Consider condensing the definitions of ‘research fatigue’, as there are unnecessary repetitions – for instance, “research fatigue’ was defined in line 47, then it was defined again in line 61, and then again in line 68. Consider positioning these definitions together in the same paragraph.

- Methods

There was no mention of how missing data was handled, or if there was any at all. This information should be included.

- Results

From Table 7, there was a strong association between fatigue and desire to drop out (OR=11.49) , and this deserves to be highlighted in the discussion, not just in the tables – especially as this was the strongest predictor of research fatigue among all the factors studied.

Also, “Socio-demographic characteristics” in Table 1 seem to be the very same characteristics re-named “Socio-demographic & socio-economic characteristics” in Table 3. Why the change in nomenclature? The former seems more succinct, and therefore more appropriate.

- Discussion

Lines 261-262 state: “Furthermore, participants who experienced research fatigue in our study were male, older, educated, and employed”. This is a repetition, as it had already been implied in Lines 254-255.

In addition, in Lines 419–427, the contrast between your findings about higher fatigue among men, and literature noting that fatigue is more prevalent in women - is interesting and could be explored further. For instance, could this reflect cultural roles or patterns in research engagement?

- Conclusion

Lines 94-96 imply that the aim of the study is to determine research fatigue “prevalence among specific contexts and identify ethical issues, and biases that could compromise the integrity and quality of data and future participation.’ If this is so, then the conclusion does not adequately reflect all of these aims.

Reviewer's summary:

In general, the manuscript does well by drawing attention to ethical and methodological challenges associated with repeated research participation, and provides evidence on its prevalence and associated factors. However, it would benefit from greater clarification of methodological details ((particularly on fatigue measurement), removing repetitions, improving flow by using clearer transitions, and adding interpretive depth in the discussion. This will significantly enhance its clarity, rigor, and impact.

**Do you want your identity to be public for this peer review?** For information about this choice, including consent withdrawal, please see our Privacy Policy

Reviewer #1: No

Reviewer #2: No

---

## [Author Response · Author response to Decision Letter 1]

9 Sep 2025

Response to reviewers

Journal Requirements:

Response

The appropriate guidelines for the authors and the corresponding author have been applied accordingly in the manuscript

 2. Please amend your authorship list in your manuscript file to include author Felishana Jepkosgei Cherop.

Response

oThis has been revised to include Felishana Cherop as the corresponding author using *

 3. Please amend the manuscript submission data (via Edit Submission) to include author Felishana Cherop.

oThis has been addressed in the system

This is well noted and no additional citation has been included

Reviewers' comments:

Reviewer's Responses to Questions

Comments to the Author

1. Is the manuscript technically sound, and do the data support the conclusions?

Reviewer #1: Yes

Reviewer #2: Yes

2. Has the statistical analysis been performed appropriately and rigorously?

Reviewer #1: Yes

Reviewer #2: I Don't Know

3. Have the authors made all data underlying the findings in their manuscript fully available?

Reviewer #1: Yesg bh

Reviewer #2: Yes

4. Is the manuscript presented in an intelligible fashion and written in standard English?

Reviewer #1: Yes

Reviewer #2: Yes

5. Review Comments to the Author

Reviewer #1: A paper of practical value to future health research studies.

A paper of practical value to future health research studies.

A paper of practical value to future health research studies.

A paper of practical value to future health research studies.

Response

Thank you very much for the positive and relevant view regarding our article

Reviewer #2: Reviewer's comments:

The authors have addressed "research fatigue" - an issue that is important yet generally under-explored. This is an important and well-conceived study that addresses a critical gap in community-engaged research ethics. However, the manuscript could be further improved by making the following adjustments:

Major Comments:

- Abstract

While the aim of the study was to determine prevalence of research fatigue and associated factors, the conclusion portion of the abstract did not report on prevalence.

Response

The abstract has been revised to include the prevalence of research fatigue and revised in the conclusion section

oThere is a high prevalence of research fatigue (56.3%) among community members in Mosoriot who have participated in repeated research, which is associated with age, gender, income source, research setting, frequency of research engagement, desire to drop out of studies, and discomfort with questions. Addressing research fatigue would enhance ethical research conduct and promote sustained community participation in research.

- Background

The background is informative but overly long. Consider condensing the definitions of ‘research fatigue’, as there are unnecessary repetitions – for instance, “research fatigue’ was defined in line 47, then it was defined again in line 61, and then again in line 68. Consider positioning these definitions together in the same paragraph.

We have revised the background to three brief paragraphs that communicate the problem and the need for the study.

oThe value of health research in promoting the health of communities cannot be understated because it addresses community and patients' needs and fosters a collaborative and ethical foundation for scientific discovery (1). Engaging communities in research ensures recognition of community priorities, values, and interests that would have intrinsic ethical importance (2). However, conducting too many research studies in one community causes research fatigue (3), raises ethical issues (4), and influences data quality (5) and future participation (6). Research fatigue is a state where individuals or groups tire of participating in research due to high-volume research projects (7), long or sensitive interviews (3,8,9), and lack of tangible benefit, often resulting to distrust (10).

oAlthough the global prevalence of research fatigue is still unknown (4), evidence shows substantial prevalence of 42% among injecting drug users (IDUs) in HIV studies in Karachi, (11) and 52% in pooled cancer studies (5). As a result, research fatigue has several undesired outcomes including research participants may become hostile (12,13), distrustful, or feel coerced, (12,18-19), hence undermining the Belmont ethical principles and guidelines for the protection of human subjects in research including respect for persons, beneficence, and justice (4,6,12,20-25). It also threatens scientific validity of research through selection and non-response biases and social desirability effects (4,18,19), which may compromise data quality and generalizability of findings(20,21,23).

oIn recognition of the presence of research fatigue in health research and insufficient data in communities that have been involved in multiple studies, there is a need to document its prevalence and ethical implications in contexts such as Kenya. This study reports findings from a community frequently engaged in multiple health research projects in Kenya.

- Methods

There was no mention of how missing data was handled, or if there was any at all. This information should be included.

Under data management and analysis, we have included a sentence indicating that there was no missing data in the collected datasets.

oWe used Statistical Package for Social Sciences (SPSS) version 21 to manage the data. There were no recorded missing data in the 327 responses….

- Results

From Table 7, there was a strong association between fatigue and desire to drop out (OR=11.49) , and this deserves to be highlighted in the discussion, not just in the tables – especially as this was the strongest predictor of research fatigue among all the factors studied.

We have included a deepened discussion on this outcome to show that it was the strongest predictor and the implication.

oThe most concerning finding was the desire to drop out of research that emerged as the strongest predictor of research fatigue. Affected participants were more than eleven times more likely to report fatigue, making withdrawal intent a critical signal of participant distress. Ethically, this indicates that autonomy, comfort, and wellbeing are at risk. Methodologically, consistent withdrawal intentions threaten threaten study validity through attrition and biased responses. Similar observations have been reported elsewhere, where repeated participation without benefit, rest or support has increased disengagement, or mistrust towards research initiatives (4).



Also, “Socio-demographic characteristics” in Table 1 seem to be the very same characteristics re-named “Socio-demographic & socio-economic characteristics” in Table 3. Why the change in nomenclature? The former seems more succinct, and therefore more appropriate.

oThe nomenclature in Table 1 and Table 3 has been aligned to reflect the same characteristics. For the reporting, we have reorganized the results under the bivariate analyses because we determined bivariate associations between research fatigue characteristics and; participant socio-demographics, research participation characteristics, understanding of ethical issues and response biases in research. Therefore, Table 1 only reports the participant demographic characteristics frequencies for those who participated in the study. Table 3 and subsequent tables under bivariate analyses, reports the association results.

- Discussion

Lines 261-262 state: “Furthermore, participants who experienced research fatigue in our study were male, older, educated, and employed”. This is a repetition, as it had already been implied in Lines 254-255.

We have amended this section to delete the repetition.

In addition, in Lines 419–427, the contrast between your findings about higher fatigue among men, and literature noting that fatigue is more prevalent in women - is interesting and could be explored further. For instance, could this reflect cultural roles or patterns in research engagement?

Thank you very much for point out this. We will consider a further exploration into this to reflect “cultural roles or patterns in research engagement”

- Conclusion

Lines 94-96 imply that the aim of the study is to determine research fatigue “prevalence among specific contexts and identify ethical issues, and biases that could compromise the integrity and quality of data and future participation.’ If this is so, then the conclusion does not adequately reflect all of these aims.

Thank you very much for this comment. We have revised the conclusion to reflect the results.

1.There was a high prevalence of research fatigue (56.3%) among community members in Mosoriot who have repeatedly engaged in health research. Research fatigue was strongly associated with socio-demographic and socio-economic factors (age, gender, education, income, research setting, and travel time to research setting), research participation characteristics (number of times in research, prolonged study duration, desire to drop out of study, being asked personal and similar questions), understanding of ethical issues(length of consenting) and response bias (answering questions in a way to protect personal information). The findings highlight highlight that research fatigue a participant burden, but also an ethical issue and a source of bias that may compromise the integrity of data and the willingness of communities to engage in future studies.



Reviewer's summary:

In general, the manuscript does well by drawing attention to ethical and methodological challenges associated with repeated research participation, and provides evidence on its prevalence and associated factors. However, it would benefit from greater clarification of methodological details ((particularly on fatigue measurement), removing repetitions, improving flow by using clearer transitions, and adding interpretive depth in the discussion. This will significantly enhance its clarity, rigor, and impact.

1.Regarding fatigue measurement, we thank the reviewer for this important point. We acknowledge that there are currently no standardized or validated measures of research fatigue in the context of health research participation. For this reason, we developed our own instrument, focusing on physical, emotional, and mental exhaustion, as well as related experiences of participation. We have clarified this in the Methods section. We also piloted and refined the tool to ensure relevance and comprehension. Nonetheless, we agree that this is a limitation of our study, and we have added this point to the Limitations section, noting the need for future work to develop validated instruments for measuring research fatigue. This is how we have responded in the methods section:

Study tool

The instrument (S1_study tool) had four major parts namely; participant socio-demographics (age, gender, marital status, sex, level of education) and Socio-economic factors included; (income, number of people in household), Types of research studies (experimental, longitudinal, and cross-sectional), research fatigue, ethical issues and biases in research. Research fatigue was assessed by asking participants to indicate if they had felt any form of fatigue exhaustion in the multiple health research studies they have participated in, namely, physical exhaustion, emotional exhaustion, and mental exhaustion. The questions were in binary response of Yes/ No. As there were no standardized or validated tools available for measuring research fatigue in the context of health research participation, we developed a structured questionnaire to capture this construct. The tool was informed by literature on participant burden and fatigue, and piloted to ensure clarity and cultural appropriateness. Although not representing a validated scale, this instrument enabled relevant data collection on research fatigue in this population…..

2. Regarding repetitions, we have removed the repetitions.

3. We have improved flow by using clearer transitions and,

4. added interpretive depth in the discussion

6. PLOS authors have the option to publish the peer review history of their article (what does this mean?). If published, this will include your full peer review and any attached files.

Do you want your identity to be public for this peer review? For information about this choice, including consent withdrawal, please see our Privacy Policy.

Reviewer #1: No

Reviewer #2: No

---

## [Decision Letter · Decision Letter 1]

23 Dec 2025

“KUCHOKA”: Investigation of Research Fatigue in Mosoriot, Kenya

PONE-D-25-02862R1

Dear Dr. Cherop,

We’re pleased to inform you that your manuscript has been judged scientifically suitable for publication and will be formally accepted for publication once it meets all outstanding technical requirements.

Kind regards,

Tülin Otbiçer Acar, PhD

Academic Editor

PLOS One

Additional Editor Comments (optional):

Reviewers' comments:

Reviewer's Responses to Questions

**Comments to the Author**

Reviewer #2: (No Response)

Reviewer #3: (No Response)

2. Is the manuscript technically sound, and do the data support the conclusions?

Reviewer #2: Yes

Reviewer #3: Yes

3. Has the statistical analysis been performed appropriately and rigorously?

Reviewer #2: I Don't Know

Reviewer #3: I Don't Know

4. Have the authors made all data underlying the findings in their manuscript fully available?

Reviewer #2: Yes

Reviewer #3: Yes

5. Is the manuscript presented in an intelligible fashion and written in standard English?

Reviewer #2: Yes

Reviewer #3: Yes

Reviewer #2: The authors have adequately addressed all the comments I raised in a previous round of review; and I feel that this manuscript is now acceptable for publication. However, there is just one more thing that may need to be included by the authors - In line 269, the authors included the phrase "....desire to drop out of the research..." - but this phrase had no citation to support it. The authors may either provide a citation for this statement, or delete it from the sentence especially since lines 285 - 287 already introduces that concept/line of discussion. Other than this, I believe the manuscript is acceptable for publication.

Reviewer #3: This topic is highly relevant given the increasing number of health-related studies conducted in resource-limited settings, and the manuscript demonstrates considerable effort in data collection and analysis. I appreciate the authors for undertaking such an important and timely study. Nevertheless, I have several concerns and questions that I believe require clarification to strengthen the manuscript and ensure its findings are fully understood by the readers.

Specific Points of Concern:

1. In the methodology section, the authors stated, “The Mosoriot community experiences diverse health-related challenges and various research projects have been conducted by different Ministry of Health and university institutions to address them.” (Ref. line 106, page 5). Additionally, in the study tool, the authors marked out health research participation characteristics, which were assessed by asking participants about the types of health research they had been involved in. (Ref. line 150, page 7). However, it seems that some critical pieces of information are overlooked. Firstly, the questionnaire does not appear to include any queries addressing the participants’ underlying health conditions (e.g., whether they were living with chronic, incurable, or manageable illnesses, or whether due to their condition, any medications they were using during participation that could have affected on their holistic health? (e.g., make the participants weak or caused any mental, physical or emotional impact on the participant’s overall health). Given that, this piece of information seems crucial since it can help to determine whether the nature of participants’ health problems have influenced their experience of research fatigue. For example, individuals with long-term or severe conditions might experience research fatigue differently compared to those with minor or temporary health issues. Secondly, it is important to know what kind of health challenges does this community suffer from and why. For instance, is it due to genetic related or environmental factors such as contaminated water, food, air etc., made this particular community prone to different health related issues? Without these details, it is difficult to determine whether research fatigue is solely a function of socio-demographic and socio-economic characteristics or it is also shaped by the participants’ health status. Therefore, I recommend that the authors clarify, if such data were collected, and if not, discuss the implications of this omission.

2. According to Table 1, socio-demographic characteristics of responders in the survey (Ref. page 9), the majority of participants were engaged in household-based research (59.9%), followed by hospital-based (32.7%) and other settings (7.3%). Whereas, Table 3 (Ref. page 11), association of research fatigue and socio-demographic and socio-economic characters, the hospital- based research is dominating (75.7%), followed by household (46.9%). I would like to ask the authors to clarify this contradictory: if household participation was the most common, why does hospital participation emerge as the stronger predictor of research fatigue?

3. The manuscript claimed that being male was associated with higher levels of research fatigue (Ref. line242, page 13) and the authors outlined this to “cultural or contextual factors unique to Mosoriot community, Kenya, however, the explanation remains vague. This finding is intriguing, since, the authors also mentioned that their study result is opposing other systematic review and meta-analysis that reported females to be more predominant than males” (Ref. line 264-5, page 15). It is advisable to give more details about what specific cultural dynamics might place greater demands on men in this community, leading to higher research fatigue? This would greatly enhance the manuscript’s comprehension.

4. Across multiple tables—including tables 3-6, the same figure (Research Fatigue Levels n = 184, 56.3%) appears repeatedly. It is unclear why this number remains constant across different variables. Since, there is no other explanation about how these numbers are obtained. I believe the readers would benefit from a clear explanation of how this number was derived and why it appears consistently across diverse categories.

5. The introduction of the abstract would benefit from additional elaboration, since the abstract is the summary of the whole paper, I suggest to mention the rationale behind the selection of this specific community as the focus of the study in this section. (Ref. line 106, page 05). Also, briefly explain the concept of “research fatigue” in the abstract to ensure clarity for the readers who may be unfamiliar with this topic.

6. The manuscript currently lacks consistency in citation style. In some sections, references are indicated numerically, while in others, the author-date format is used (Ref. lines 50 and 87, pages 3–4). I strongly advise the authors to follow a single citation style and apply it uniformly throughout the manuscript.

7. The study tool section contains unnecessary redundancies. Specifically, the sentence “The questions were binary response of Yes/No” is repeated four times within a single paragraph (Ref. lines 149–159, page 7). Hence, I suggest to simplify this by stating once, at the beginning, that all the questions were asked in a binary Yes/No format. This will eliminate repetition and improve legibility for the readers.

**Do you want your identity to be public for this peer review?** For information about this choice, including consent withdrawal, please see our Privacy Policy

Reviewer #2: **Yes: ** Omolayo T. Umaru

Reviewer #3: **Yes: ** Arezou Ahmadi R.A

---

## [Editor Report · Acceptance letter]

PONE-D-25-02862R1

PLOS One

Dear Dr. Cherop,

I'm pleased to inform you that your manuscript has been deemed suitable for publication in PLOS One. Congratulations! Your manuscript is now being handed over to our production team.

Kind regards,

on behalf of

Dr. Tülin Otbiçer Acar

Academic Editor

PLOS One